# The Mediating Role of Employee Engagement in the Relationship between Flexible Work Arrangements and Turnover Intentions among Highly Educated Employees in the Republic of Serbia

**DOI:** 10.3390/bs13020131

**Published:** 2023-02-05

**Authors:** Dimitrije Gašić, Nemanja Berber

**Affiliations:** Faculty of Economics in Subotica, University of Novi Sad, 21102 Novi Sad, Serbia

**Keywords:** flexible work arrangements, employee engagement, turnover intentions, attitudes of employees, Republic of Serbia

## Abstract

The main objective of this research was to investigate the relationship between flexible working arrangements (FWA) and turnover intentions (TI), as well as the mediation effect of employee engagement (EE) in the relationship between flexible working arrangements and turnover intentions. The main research question is: what is the nature of the effect of flexible working arrangements on employees’ turnover intention, and the role of employee engagement in this relationship? The methodology of the paper consists of theoretical (literature review) and empirical parts (field research). The empirical research was performed on a sample of 514 highly educated employees from service sector organizations that operate in the Republic of Serbia. Sample collection lasted from January to October 2022, via Google Forms. The proposed relationships were tested by using the PLS-SEM method, with the application of the SmartPLS software. The main findings of the research are that there are direct positive effects of flexible work arrangements and employee engagement on turnover intentions, and that employee engagement has an indirect effect on the relationship between flexible work arrangements and turnover intentions. A partial mediation was found. Employees who are offered flexible work arrangements decrease their turnover intentions when they are more engaged at work.

## 1. Introduction

Modern organizations face various challenges such as increasingly strong competition, the rapid development of information and communication technologies, rapid globalization, enhanced digitalization, and economic, political, energy, health (COVID-19 pandemic), and other crises. In such circumstances, organizations need to adapt their business patterns and find a strategy with the aim not only to survive but also to achieve a leadership position in the market [1,2,3,4]. Additionally, the labor market has changed, and one of the most important changes is the presence of Y and Z generations in the labor market, who are quite different from previous generations in terms of their attitudes toward business, the balance between their job and private life, authority, environment, career development, etc. [5,6,7]. Consequently, as a response to all of the above-mentioned, organizations are forced to redesign work and implement different forms of flexible work arrangements such as home-based work, teleworking, reduced hours, weekend work, shift work, a compressed working week, etc. [8,9,10,11]. Flexible work arrangements allow employees to maintain a balance between their job and private life, and to choose how, where, and when they will execute their business tasks [12,13,14,15].

The main objective of this research is to examine the relationship between flexible work arrangements and turnover intentions, as well as the mediating effect of employee engagement. The sample was created using an electronic Google Forms questionnaire where employees in the positions of professional workers or managers belonging to the service sector in the territory of the Republic of Serbia had the opportunity to answer at any time. After collecting an adequate number of respondents (514 employees), the authors coded the data and then conducted a PLS-SEM analysis to determine the relationships between the observed variables. Flexible work arrangements represent an independent variable, while employee engagement and turnover intentions represent dependent variables. Moreover, it should be emphasized that employee engagement and turnover intentions represent employees’ attitudes, and the attitudes are significant predictors of the future behavior of employees [16]. They represent employees’ cognitive, affective, and behavioral reactions to different aspects of work [17].

The research consists of four parts, the first part refers to the theoretical background of the research, where the authors explain the research variables (flexible work arrangements, employee engagement, and turnover intentions) and emphasize their importance for the research. This is followed by an overview of previous research in the area, and the development of hypotheses. The methodology of the research is presented in the second section. The authors present the questionnaires used, data analysis procedures, as well as the sample. The third part is related to the empirical part of the research, in which the authors used the SmartPLS and IBM SPSS Statistics programs for data processing and performed PLS-SEM analysis to test the proposed relationships. Finally, the authors provide conclusions regarding the entire research, theoretical and empirical implications, research limitations, and proposed suggestions for future research.

## 2. Theoretical Background

### 2.1. Flexible Work Arrangements, Employee Engagement, and Turnover Intentions

Due to various internal and external factors that have a direct impact on business, modern organizations are using flexible work arrangements. Companies have modified the traditional way of organizing work and today there are different flexible work arrangements in usage, such as weekend work (employees can work during the weekend and/or can work in two shifts), shift work (different employees work at different times in the workplace, so the organization can work longer than 8 h (16 h, or even 24 h if they work in three shifts)), overtime (employees have the possibility to work additional hours beyond those stipulated in their employment contracts), flexi-time (the employee can choose when he/she will start and end their work, accompanied by agreement with the management of the company), home-based work (employees have the opportunity to work from their home, outside their office in the company), teleworking (employees work out of the office, but a strong internet connection and computer device are needed), a compressed working week (it means that instead of five days of 8 h, compression is performed for fewer working days but more working hours, for example, 40 working hours are actually “condensed” into four working days, by extending the working day to 9 or 10 h), job share (part-time employees can share a full-time job), part-time jobs (employees work less than the regular 40 h working week), fixed-term contracts (employees work under employment contracts with a fixed duration), temporary/casual work (employment is offered temporarily), annual hours contracts (a predetermined number of working hours per year) [8,9,11]. 

Flexible work arrangements bring mutual benefits for both employers and employees. Both parties agree when, where, and how employees will work with the aim of meeting the needs of the company [18]. Organizations that implement flexible work arrangements in their business are aware that changing work patterns in the direction of flexibility can positively affect a better balance between the work and private life of employees [14,15]. For example, teleworking is one of the most common flexible work patterns around the world and is expected to become more prevalent in the future. However, according to Kossek and Lautsch [19], as a type of flexible work arrangement, teleworking was not used widely before the outbreak of the COVID-19 pandemic. In addition to teleworking, working from home became the “new norm” for many organizations during the COVID-19 pandemic. Millions of people worked from their homes and/or even remotely, from different safe places, by using internet connections and electronic devices (desktop computer, laptop, tablet, phone, etc.), to work with their colleagues [20]. Buruck et al. [21] investigated the relationship between contextual work-related factors, where they found, among other results, that work flexibility has a negative effect on the burnout dimension of emotional exhaustion, and that the possibility of taking a day off or using other flexible arrangements reduces work–life conflicts.

Employee attitudes are significant predictors of future behavior [16], and represent the cognitive, affective, and behavioral reactions of employees to different aspects of work [17]. The goal of applying flexible work arrangements is flexibility, employee well-being, and increased success, and accordingly, it is necessary to investigate relationships with employees’ attitudes related to the work. The attitudes of employees that the authors investigate in this study are employee engagement and turnover intentions. Employee engagement refers to a positive, fulfilling work-related state of mind characterized by vigor, commitment, and absorption. Vigor refers to a high level of energy and mental resilience during work. Commitment refers to being strongly involved in one’s work and experiencing a sense of significance, enthusiasm, inspiration, pride, and challenge. Absorption is characterized by complete concentration and preoccupation with work, where time passes quickly and a person has difficulty separating from work [22]. Turnover intentions are defined as the conscious and deliberate intention of the individual to leave the job, and they are described as the last in a series of cognitions that precede withdrawal from the job [16,23]. 

### 2.2. Relations between Flexible Work Arrangements, Employee Engagement, and Turnover Intentions

Investigation of the direct relations between flexible work arrangements, employee engagement, and turnover intentions, as well as the role of employee engagement in the relationship between flexible work arrangements and turnover intentions has been the theme of much of the previous research. The main theoretical background for these relations can be found in the social exchange theory [24]. The main idea of the social exchange theory is that “positive behavior of one person (sender) to another (receiver) in an interdependent relationship would create the potential for the receiver to feel obligated to reciprocate with returned positive behavior” [7]. Accordingly, if employees perceive the practices of employers as positive, they will show positive work attitudes and behaviors. Conversely, if employees perceive employers’ practices as unfair or negative, they will show negative organizational behavior and attitudes. We can expect that flexible work arrangements would have positive effects on both employee engagement and turnover intentions, as types of employee attitudes, in terms of decreasing turnover and increasing employee engagement. 

Drawing from the social exchange theory, employees who perceive flexible work arrangements positively, in terms of all the benefits that flexible work arrangements can bring to an employee, could feel more engaged at work. Employees who have the possibility to use flexible work arrangements achieve some greater autonomy and control over their job, and enhance their work–life balance, and they can feel a higher level of energy and mental resilience during work, a stronger involvement in their work, and experience a sense of significance, enthusiasm, inspiration, pride, and challenge, and higher concentration and preoccupation with work. The theoretical linkage between flexible work arrangements and employee engagement lies in the discretion afforded to employees as to how work is completed and how workers can achieve some mechanisms of control and autonomy [25]. Previous research also investigated these relations. In a study of Ugargol and Patrick [26] on a sample of 504 employees in IT companies in Bengaluru (India), the study indicated that flexible work arrangements are positively related to employee engagement. The main objective of the Basheer et al. [27] study was to find the role of employee engagement in improving the effect of spiritual intelligence, emotional intelligence, and flexible work arrangements on employee loyalty in the PROTON automotive industry of Malaysia. The results of the study revealed that spiritual intelligence, emotional intelligence, and flexible work arrangements have a positive relationship with employee loyalty. Furthermore, employee engagement has been found to be a vital factor in increasing employee loyalty through spiritual intelligence, emotional intelligence, and flexible work arrangements. Weideman and Hofmeyr [28] found a positive relationship between flexible work arrangements and employee engagement as well as various employee engagement constructs found in the literature, with the most prominent finding showing the positive impact of flexible work arrangements on employee well-being. Gašić and Berber [11] performed a PLS-SEM on a sample of 219 employees in the Republic of Serbia and determined that flexible work arrangements have a direct positive effect on employee engagement. Based on the above-mentioned, the authors proposed the first hypothesis:

**Hypothesis** **1** **(H1).**
*Flexible work arrangements are positively related to employee engagement.*


Regarding turnover intention, this attitude could be decreased because employees, when they have the opportunity to use flexible work arrangements, can have more control and autonomy regarding their business and private life, improve their work–life balance, and on that basis, feel more satisfied in the job and have a desire to stay in a specific company. If employees perceive flexible work arrangements as a positive business practice, it is expected that they will have positive work attitudes, such as job satisfaction, employee commitment, and engagement, and on that basis, that they will not try to avoid their work and leave their organization. The results of previous research [29,30,31] proved this proposition. According to McNall, Masuda, and Nicklin [29], based on hierarchical regression analysis on a sample of 220 employees, work-to-family enrichment mediated the relationship between flexible work arrangements, job satisfaction, and turnover intentions, even after controlling for gender, age, and other demographic variables. Being able to implement flexible work arrangements such as flexible working hours and a compressed work week helps employees experience greater work-to-home enrichment, which is associated with higher job satisfaction and lower turnover intentions. Regarding the research of Azar et al. [30] on flexible work arrangements and organizational outcomes, based on a sample of 289 employees, the results showed that job satisfaction and work–life conflict mediated the relationship between flexible work arrangements and turnover intentions, controlling for gender, age, marital status, and other demographic variables. Bontrager, Clinton, and Tyner [31] emphasize that flexible work arrangements address work–life balance issues, and this was especially so during the COVID-19 pandemic, and that flexible work arrangements can be used to reduce turnover intentions and “facilitate employee development through work-life balance programs”. In the context of Serbia, Gašić and Berber [11] performed a PLS-SEM on a sample of 219 employees and found that flexible work arrangements have a direct positive effect on turnover intentions, more precisely, a negative relation to turnover intentions. Based on the mentioned theory propositions, and previous research results, the authors of this research proposed the second hypothesis:

**Hypothesis** **2** **(H2).**
*Flexible work arrangements are negatively related to turnover intentions.*


As well as direct relations, it is obvious that other attitudes of employees can have some indirect effects on FWAs-TI relations. This is tackled in the previous part of the research since the authors stated that employees usually feel a kind of satisfaction when they use different FWAs, and then estimate the desire to stay or leave the organization. However, satisfaction is not the only attitude that is related to a desire to stay in a company. Higher engagement at work leads to loyalty and a stronger attachment to an organization, and this results in lower levels of employee turnover intentions [32,33,34]. There have only been a few researches in the past that have addressed this indirect relationship. Tsen et al. [35] emphasized that organizational commitment and work–family conflicts are significant mediators in all models presented in the paper. In models related to social exchange theory, all flexible work arrangements lead to increased organizational commitment before decreasing turnover intentions, implying the beneficial outcomes of flexible work arrangements. Additionally, Tsen et al. [36] indicated that perceived job independence plays a moderator role in the relationship between flexible work arrangements and turnover intentions. Yamin and Pusparini [37] examined the effect of flexible work arrangements and perceived organizational support (POS) on employee performance through employee engagement. The findings of this study showed that flexible work arrangements and POS have a positive and significant impact on employee engagement and employee performance. The study also found that employee engagement had a positive and significant impact on employee performance. Basheer et al. [27] found that employee engagement is a vital factor (mediator) in increasing employee loyalty through spiritual intelligence, emotional intelligence, and flexible work arrangements. When employees are loyal to their organizations, they probably will not leave them. Based on the aforementioned, flexible work arrangements have been found to be positively related to work-related attitudes such as engagement, commitment, and job satisfaction. Moreover, it has been found that flexible work arrangements are negatively related to turnover intentions. The authors of this research propose that employee engagement mediates the relationship between flexible work arrangements and turnover intentions and that employees who perceive their flexible work arrangements better have a lower level of turnover intentions when they are more engaged on the job. Based on the theoretical background of the research and the results of other authors who examined the proposed relationships, the authors proposed the third hypothesis:

**Hypothesis** **3** **(H3).**
*Employee engagement has a positive mediation effect in the relationship between flexible work arrangements and turnover intentions.*


## 3. Methodology

### 3.1. The Questionnaire

Data collection was performed through an electronic Google Forms questionnaire, where professional workers and managers in the service sector could answer questions at any time. The questionnaire consists of four parts. The first part refers to the socio-demographic and organizational characteristics (gender, age, education, size of the organization (SMEs or large), and belonging to the public or private sector). The second part refers to the question about Flexible Work Arrangements where the employees had the opportunity to answer 11 questions (see Table 1). Flexible working arrangements represent a way of doing business that allows employees to adequately organize their working time, business and private obligations (work–life balance), activities related to solving business tasks, as well as reaching defined goals [38,39]. If employees have the opportunity to use some of the forms of flexible work arrangements, they will be more productive and efficient when solving business tasks, and this will have a favorable effect on increasing the success of the organization, so we can conclude that there is mutual benefit for both the employee and the employer. The 11-item measure of flexible work arrangements based on Albion [40] was used as the independent variable. Respondents answered on a 5-point Likert-type scale, from 1 to 5 [41]. High scores indicate a higher positive perception of employees toward flexible work arrangements.

The third part refers to Employee Engagement where the employees had the opportunity to answer 9 questions (see Table 2). Employee engagement refers to a positive, fulfilling work-related state of mind characterized by vigor, dedication, and absorption [22]. Employee engagement was measured with the 9-item short version of the Utrecht work engagement scale (UWES; [42]). All used items are presented in Table 2. Respondents answered on a 5-point Likert-type scale, from 1 to 5. High scores indicate higher levels of work engagement.

The fourth part refers to Turnover Intentions where the employees had the opportunity to answer 4 questions (see Table 3). Turnover intention is defined as the conscious and deliberate intention of the individual to leave the job and is described as the last in a series of cognitions that precedes withdrawal from the job [23]. Chen and Francesco’s [43] four-item turnover intentions measure was included. All indicators (questions) are presented in Table 3. Respondents answered on a 5-point Likert-type scale, from 1 to 5. High scores indicate higher levels of turnover intentions of employees.

If the questionnaire contains a reverse score at the end of the question, it means that the authors made the necessary changes before the analysis.

### 3.2. The Sample

After creating the electronic questionnaire, the authors moved on to the next step related to sample collection. The sample was made up from employees from the territory of the Republic of Serbia who belong to the service sector, work in positions such as professional workers or managers, and who are highly educated. Based on the data from the Statistical Office of the Republic of Serbia (2022), there were 2,848,800 employed persons in the Republic of Serbia. Because flexible work arrangements are offered to a greater extent to employees with higher education [44] and in higher positions (white collars), the authors of this paper decided to investigate employed persons that are highly educated. There were 27.7% of employees in Serbia who possessed higher education degrees (which made 789,118 employees). The authors used a sample size formula developed by Cochran (1977) to find out the necessary sample size. Based on Cochran’s calculation, with the most commonly applied significance level of 5% and a confidence interval of 95%, an appropriate sample in the research was considered to be a sample consisting of a minimum of 385 highly educated employees in the Republic of Serbia.

Sample collection lasted from January to October 2022; during that time, a total of 582 responses were gathered. After the data check and preparation (exclusion of outliers and incomplete questionnaires), only 514 respondents who filled out the questionnaire were taken into account. The structure of the sample is shown in Table 4.

Based on the presented Table 4, females made up 56.6% of the population, while the rest were male, 223 (43.4%). When comparing the age of respondents, the largest number of respondents were between 25 and 35 years of life, 211 (41.1%), while the smallest number of them were over 55 years, 28 (5.4%). By comparing the level of education of employees, the largest number of them held master’s degrees, 188 (36.6%), while the smallest share of respondents had completed three-year vocational studies, 44 (8.5%). The largest number of them worked in SMEs, 357 (69.5%), while the rest of them worked in large organizations, 157 (30.5%). Comparing public and private sector affiliation, the private sector made up 61.5% of the sample, while the rest belonged to the public sector, 198 (38.5%).

### 3.3. Data Analysis

The authors used Partial Least Squares Structural Equation Modeling (PLS-SEM) to test the proposed model. PLS-SEM is a method based on the analysis of complex interrelated relationships between constructs and indicators [45] (p. 2). PLS path models can be considered to have two sets of linear equations as follows: measurement model (outer model) and structural model (inner model). While the outer model specifies the relationship between a construct and its observed indicators, the inner model specifies the relationships between the constructs. PLS-SEM was used for analysis as it is based on a series of ordinary least square (OLS) regressions, has only minimum demands with respect to sample size, is capable of achieving higher levels of statistical power, and is capable of providing more stable and robust results [46]. PLS-SEM has gained widespread esteem across various industries, including human resource management. When evaluating the structural model, PLS-SEM has the advantage of studying latent constructs through path analysis and emphasizing the explanation of variance in dependent variables [47]. The estimation of PLS path model was performed in four steps. The first step was the creation of the iterative algorithm that involves identifying the composite scores for each of the constructs. The second was a correction for attenuation, which models the constructs as factors. The third step was the estimation of the parameter, and the fourth step was bootstrapping, which tests the inferences. Regarding the path analysis, the measurement model connections between latent and observable variables were tested by using indicator reliability and construct reliability, and for validity, convergent and discriminant validities, while the structural model was tested by application of the bootstrapping procedure based on 5000 subsamples. 

## 4. Result and Discussion

The study assessed the direct effects of flexible work arrangements on the dependent variables (employee engagement and turnover intentions) and indirect effects through the mediator (employee engagement). The authors first tested the formative higher-order construct flexible work arrangements.

The formative construct in the model was tested by analyzing the outer weight as well as the significance through t-statistics and the *p*-value. Multicollinearity analysis was also performed by the variance inflation factor (VIF).

Based on the values in Table 5, which represent a part of the outer model for the formative construct, the authors conclude that all coefficients of the path are positive and significant (*p*-values < 0.05). Formative construct flexible work arrangements consists of two variables, FWAs related to family issues and FWAs related to job issues.

The multicollinearity was tested by the variance inflation factor (VIF). Based on the obtained results in Table 6, there is no multicollinearity between the formative constructs because the values of the VIFs are less than 3.3 [48]. Based on the presented results, it can be concluded that the formative second order construct is reliable and valid.

To test reflective constructs in the model, the reflective indicator loadings, internal construct reliability, convergent validity, and discriminant validity were analyzed [7]. Regarding reflective indicator loadings, factors loadings between 0.4 and 0.7 should be retained only if their removal does not have an impact on AVE and composite reliability, but the eligible factor loadings should be above 0.708 [49]. The items FWA2R, FWA4R, FWA8R, FWA11, and ENG9 had to be removed from further analysis because their load level was lower than acceptable. Figure 1 shows the used items (questions) with the accepted loadings (see Figure 1).

Table 7 represents the indicator and construct reliability test. The authors tested Cronbach’s alpha, composite reliability, and average variance extracted (AVE). Cronbach’s alpha values were 0.681, (absorption), 0.772 (flexible work arrangements job), 0.819 (flexible work arrangements family), 0.834 (dedication), 0.846 (vigor), and 0.861 for turnover intentions. The value of the composite reliability of constructs ranged from 0.862 (absorption), 0.868 (flexible work arrangements job), 0.882 (flexible work arrangements family), 0.901 (dedication), 0.906 (turnover intentions), to 0.907 for vigor. Convergent validity was assessed by testing the average variance extracted (AVE), ranging from 0.652 (flexible work arrangements family), 0.687 (flexible work arrangements job), 0.707 (turnover intentions), 0.751 (dedication), 0.758 (absorption) to the highest value, for vigor (0.756). Based on the values shown in Table 3, we conclude that all criteria are met for all three observed analyses.

Discriminant validity was assessed by cross-loading indicators, the Fornell–Larcker Criterion, and heterotrait–monotrait, HTMT [52].

Authors Grubor et al. [53] (p. 310), emphasize that external loadings of indicators of the corresponding construct should be higher than all cross-loadings with other constructs. Based on the cross-loading assessment shown in Table 8, the external loadings of the indicator of the corresponding construct are higher than all the cross-loadings with other constructs and we can conclude that the criterion is met.

According to the Fornell–Larcker criterion, the square root of the AVE of a latent variable must have a higher value than all correlations with other latent variables [54]. Table 9 demonstrates that the criterion for discriminant validity is satisfied because the value of the square root of AVE is higher than all values below for each variable, respectively [11]. 

HTMT ratio values below 0.9 indicate that the defined components are sufficiently different from each other; it means that they describe different phenomena [55]. The results presented in Table 10 show that all values are below 0.9, so it can be concluded that the discriminant validity criterion is met.

Based on the data in Table 11, all variance inflation factor (VIF) values are below 3.3 [48], so we conclude that the measurement of the structural model is reliable and valid in terms of the multicollinearity test.

In order to test the observed relations (structural model), a bootstrapping analysis was performed. Subsamples are randomly drawn observations from the original data set (with replacement). The subsample is then used to estimate a PLS path model. The process is repeated until a large number of random subsamples (for example 5000) have been generated. Estimates from bootstrap subsamples are used when determining standard errors for PLS-SEM results. With this information, t-values, *p*-values, and confidence intervals are calculated to assess the significance of the PLS-SEM results. The results obtained on the basis of the bootstrapping analysis are shown in the following table and graph on the basis of which we can check the proposed hypotheses [56].

The data in Table 12 indicate a positive statistically significant relationship between flexible work arrangements and employee engagement (β = 0.209; t = 4.918; *p* = 0.000). Based on the presented results, hypothesis H_1_ is confirmed. Additionally, there is a negative statistically significant relationship between flexible work arrangements and turnover intentions (β = −0.197; t = 4.195; *p* = 0.000), which confirms hypothesis H_2_. Finally, the author tested the mediating role of employee engagement in explaining the relationship between flexible work arrangements and turnover intentions. There is a partial mediation since the indirect effect of flexible work arrangements on turnover intentions through employee engagement is significant (β = −0.110; t = 4.666; *p* = 0.000). Based on the presented results, hypothesis H_3_ is confirmed too. These relations are presented in Figure 2.

## 5. Conclusions

The present research reveals relations between the modern work design strategy, flexible work arrangements, employee engagement, and turnover intentions, as attitudes that have strong effects on employee behavior in a company. The research successfully confirmed all three hypotheses, i.e., flexible work arrangements have positive effects on employee engagement, and negative effects on turnover intentions, and employee engagement has a positive partial mediation effect in the relationship between flexible work arrangements and turnover intentions.

The practical implications of the research are in the potential of flexible work arrangements to increase employee engagement and reduce turnover intentions in companies for their highly educated employees. Due to the insufficient number of researches dealing with the mediating role of employee engagement between flexible work arrangements and the turnover intentions, the authors decided to examine the given relations and determine the effects. Moreover, research is of great importance for almost all companies in modern business because there is a lack of talent and a high level of stress [57] at work, as well as a high level of turnover intentions, which represent one of the most important problems faced by HRM in organizations. Companies around the world are faced with various internal and external factors such as digitization and globalization, and energy, health (COVID-19 pandemic), and political (war) crises, as well as the lack of skills and abilities of employees to perform their business activities [11]. These effects have a direct impact on their business and survival in the market, and consequently, companies have to redesign their business in order to manage their human potential more efficiently and effectively. One of the ways for a company to achieve a competitive advantage and improve its success is through the application of flexible work arrangements [7]. Therefore, the results of this research can serve as a starting point for creating strategies and actions for the implementation of flexible work arrangements to lead to positive employee attitudes such as work engagement and the intention to stay in a company. This is even more important for highly educated employees, who usually cover professional and managerial positions in companies.

Regarding the theoretical implications, the present research is one of the first to deal with the themes of flexible work arrangements, employee engagement, and turnover intentions in the Serbian context, with this specific methodology (questionnaires and PLS-SEM). We tested and confirmed all questionnaires, which is important for future research and the reliability of the findings. Theoretical implications also lie in an increased understanding of the effects of employee engagement on the relationship between flexible work arrangements and turnover intentions, which is very important bearing in mind behavioral economics, where the psychological dimension of human behavior is found to be one of the most important factors for decision making [58]. In this study, the effects of mediation were tested, and the results indicated that employee engagement can serve as a mediator in the investigated relationship, which is in line with previous research, such as that of Basheer et al. [27], who found that employee engagement is a mediator in the relationship between FWA and employee loyalty, where employee loyalty is related to a lower intention to leave the organization and a reduction in the actual turnover rate.

This study has potential limitations. One is related to the sample size. According to the data of the Republic Institute of Statistics in 2021, the average number of higher educated employees was 789,118. This sample consists of only 514 employees. Considering the size of the target population in the survey for the year 2021, and in accordance with the sample size formula developed by the authors Cochran (1977), we conclude that for the most commonly applied significance level of 5% and confidence interval of 95% [59], an appropriate sample in the research can be considered a sample consisting of a minimum of 385 employees in the Republic of Serbia. Therefore, the sample consisting of 514 highly educated employees can serve as an appropriate one for this level of analysis. In addition to the sample size, the authors did not use control variables in the analysis of the model, which could give interesting results. Therefore, the inclusion of demographic and organizational-level variables in the model, as moderators or mediators, may reveal new results, and this is a recommendation for future research. 

## Figures and Tables

**Figure 1 behavsci-13-00131-f001:**
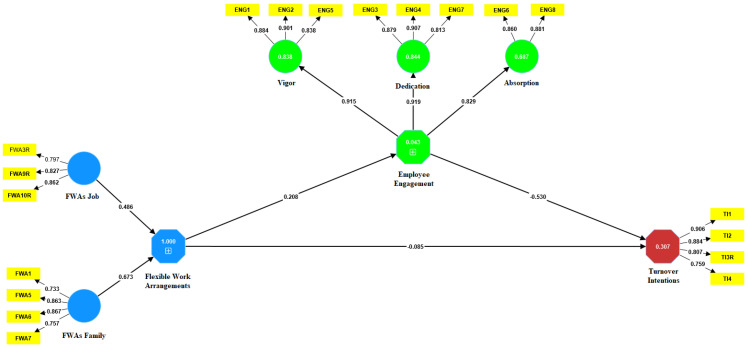
Path coefficient estimates. Source: The authors’ research.

**Figure 2 behavsci-13-00131-f002:**
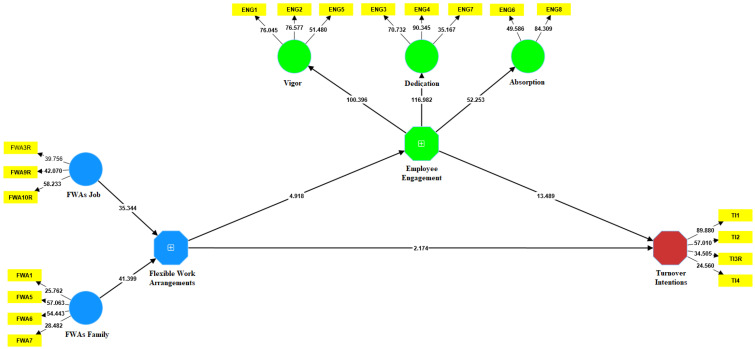
The path model with bootstrapping result. Source: The authors’ research.

**Table 1 behavsci-13-00131-t001:** Questions about the independent variable—Flexible Work Arrangements (2nd part).

Variable—Flexible Work Arrangements	Items	Questions
Flexible Work Arrangements—Job	FWAR2	I cannot afford the loss of pay associated with most Flexible Work Arrangements (R)
	FWAR3	Flexible Work Arrangements don’t suit me because they tend to make me feel disconnected from the workplace (R)
	FWAR4	Working shorter hours would negatively impact on my career progress within the organization (R)
	FWAR8	People at my workplace react negatively to people using Flexible Work Arrangement (R)
	FWAR9	People using Flexible Work Arrangements usually have less commitment to their work roles (R)
	FWAR10	People using Flexible Work Arrangements often miss important work events or communications, such as staff meetings, training sessions, important notices, etc. (R)
	FWA11	I would not be able to do paid work at all if I could not use Flexible Work Arrangements
Flexible Work Arrangements—Family	FWA1	Flexible Work Arrangements help me balance life commitments.
	FWA5	Working more flexible hours is essential for me to attend to family responsibilities.
	FWA6	Flexible Work Arrangements are essential for me to participate in family and social events.
	FWA7	Flexible Work Arrangements enables me to focus more on the job when I’m at the workplace.

Source: Albion [40].

**Table 2 behavsci-13-00131-t002:** Questions about the dependent variable—Employee Engagement (3rd part).

Variable—Employee Engagement	Items	Questions
Vigor	ENG1	At my work, I feel bursting with energy
	ENG2	At my job, I feel strong and vigorous
	ENG5	When I get up in the morning, I feel like going to work
Dedication	ENG3	I’m enthusiastic about my job
	ENG4	My job inspires me
	ENG7	I’m proud of the work that I do
Absorption	ENG6	I feel happy when I’m working intensely
	ENG8	I’m immersed in my work
	ENG9	I get carried away when I’m working

Source: Schaufeli, Bakker, and Salanova [42].

**Table 3 behavsci-13-00131-t003:** Questions about the dependent variable—Turnover Intentions (4th part).

Variable—Turnover Intentions	Items	Questions
Turnover Intentions	TI1	I often think about leaving my current job
	TI2	Maybe next year I will leave my current company and start working for someone else
	TI3R	I plan to stay in this company for a longer time to develop my career (R)
	TI4	I probably won’t have a bright future if I stay in this company

Source: Chen and Francesco [43].

**Table 4 behavsci-13-00131-t004:** Sample characteristics.

Gender	Number	Percent
Male	223	43.4
Female	291	56.6
Total	514	100
**Age structure**	**Number**	**Percent**
Less than 25	70	13.6
25–34	211	41.1
35–44	138	26.8
45–55	67	13.0
More than 55	28	5.4
Total	514	100
**Level of education**	**Number**	**Percent**
Three-year vocational studies	44	8.5
Bachelor’s degree	165	32.1
Master’s study	188	36.6
Ph.D.	117	22.8
Total	514	100
**Size of the company**	**Number**	**Percent**
SMEs	357	69.5
Large	157	30.5
Total	514	100
**Belonging to the public or private sector**	**Number**	**Percent**
Public	198	38.5
Private	316	61.5
Total	514	100

Source: The authors’ research.

**Table 5 behavsci-13-00131-t005:** Analysis of the formative construct of the external model.

Relationship	Outer Weight	Standard Deviation	T-Statistics	*p*-Values
Flexible Work Arrangements Family → Flexible Work Arrangements	0.673	0.016	41.399	0.000
Flexible Work Arrangements Job → Flexible Work Arrangements	0.486	0.014	35.344	0.000

Source: The authors’ research.

**Table 6 behavsci-13-00131-t006:** Variance inflation factor—VIF.

Formative Construct—Flexible Work Arrangements	Variance inflation factor—VIF	
	Value	Criterion
Flexible Work Arrangements Family	1.293	VIF < 3.3 [48]
Flexible Work Arrangements Job	1.293	

Source: The authors’ research.

**Table 7 behavsci-13-00131-t007:** Indicator reliability and construct reliability and validity.

Variables	Second Order	Cronbach’s Alpha		Composite Reliability		AVE	
		Value	Criterion	Value	Criterion	Value	Criterion
Flexible Work Arrangements	FWAs Family	0.819	**>0.6** [50]	0.882	**>0.7** [49]	0.652	**>0.5** [51]
	FWAs Job	0.772		0.868		0.687	
Employee Engagement	Vigor	0.846		0.907		0.765	
	Dedication	0.834		0.901		0.751	
	Absorption	0.681		0.862		0.758	
Turnover Intentions		0.861		0.906		0.707	

Source: The authors’ research.

**Table 8 behavsci-13-00131-t008:** Discriminant validity (cross-loadings).

Variable Name	Second Order	Items	EE			FWAs		TI
			Vigor	Dedication	Absorption	FWAs—Family	FWAs—Job	Turnover Intentions
Employee Engagement	Vigor	ENG1	**0.884**	0.574	0.509	0.222	0.204	−0.366
		ENG2	**0.901**	0.653	0.541	0.202	0.193	−0.413
		ENG5	**0.838**	0.712	0.665	0.117	0.155	−0.494
	Dedication	ENG3	0.736	**0.879**	0.575	0.108	0.173	−0.454
		ENG4	0.67	**0.907**	0.594	0.068	0.143	−0.495
		ENG7	0.509	**0.813**	0.572	0.115	0.08	−0.494
	Absorption	ENG6	0.557	0.54	**0.86**	0.154	0.105	−0.301
		ENG8	0.587	0.621	**0.881**	0.134	0.068	−0.382
Flexible Work Arrangements	FWAs—Family	FWA1	0.211	0.163	0.166	**0.733**	0.409	−0.189
		FWA5	0.131	0.06	0.123	**0.863**	0.347	−0.139
		FWA6	0.15	0.065	0.117	**0.867**	0.337	−0.064
		FWA7	0.169	0.073	0.127	**0.757**	0.446	−0.085
	FWAs—Job	FWA10R	0.167	0.139	0.112	0.344	**0.862**	−0.186
		FWA3R	0.172	0.08	0.043	0.464	**0.797**	−0.157
		FWA9R	0.181	0.17	0.093	0.369	**0.827**	−0.149
Turnover Intentions		TI1	−0.483	−0.526	−0.373	−0.124	−0.154	**0.906**
		TI2	−0.364	−0.427	−0.322	−0.109	−0.128	**0.884**
		TI3R	−0.431	−0.493	−0.366	−0.149	−0.175	**0.807**
		TI4	−0.342	−0.393	−0.241	−0.107	−0.216	**0.759**

Source: The authors’ research.

**Table 9 behavsci-13-00131-t009:** Discriminant validity—Fornell–Larcker criterion.

	Absorption	Dedication	FWAs Family	FWAs Job	Turnover Intentions	Vigor
Absorption	**0.871**					
Dedication	0.668	**0.867**				
FWAs Family	0.165	0.111	**0.807**			
FWAs Job	0.099	0.155	0.476	**0.829**		
Turnover Intentions	−0.394	−0.553	−0.147	−0.198	**0.841**	
Vigor	0.658	0.743	0.204	0.210	−0.488	**0.874**

Source: The authors’ research.

**Table 10 behavsci-13-00131-t010:** Discriminant validity—heterotrait–monotrait (HTMT).

	Absorption	Dedication	FWAs Family	FWAs Job	Turnover Intentions	Vigor
Absorption						
Dedication	0.887					
FWAs Family	0.222	0.137				
FWAs Job	0.139	0.192	0.598			
Turnover Intentions	0.505	0.649	0.175	0.246		
Vigor	0.861	0.874	0.249	0.260	0.562	

Source: The authors’ research.

**Table 11 behavsci-13-00131-t011:** Collinearity statistics (variance inflation factor—VIF).

Variables Name	Second Order	Items	Variance Inflation Factor—VIF	
			Value	Criterion
Flexible Work Arrangements	FWAs Job	FWA1	1.413	VIF < 3.3 [48]
		FWA5	2.724	
		FWA6	2.782	
		FWA7	1.472	
	FWAs Family	FWA3R	1.404	
		FWA9R	1.828	
		FWA10R	1.943	
Employee Engagement	Vigor	ENG1	2.689	
		ENG2	2.959	
		ENG5	2.571	
	Dedication	ENG3	2.124	
		ENG4	2.776	
		ENG7	1.679	
	Absorption	ENG6	1.691	
		ENG8	1.856	
Turnover Intentions		TI1	3.283	
		TI2	3.113	
		TI3R	1.707	
		TI4	1.653	

Source: The authors’ research.

**Table 12 behavsci-13-00131-t012:** Values of mean, standard deviation, t-statistics, and *p*-values of the structural model.

Relationship	β	Standard Deviation	t-Statistics	*p*-Values	Hypothesis
Flexible Work Arrangements → Employee Engagement	0.209	0.042	4.918	0.000	H_1_: Accepted
Flexible Work Arrangements → Turnover Intentions	−0.197	0.047	4.195	0.000	H_2_: Accepted
Flexible Work Arrangements → Employee Engagement → Turnover Intentions	−0.11	0.024	4.666	0.000	H_3_: Accepted

Source: The authors’ research.

## Data Availability

The data presented in this study are available on request from the corresponding author.

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
