# Peer review of "The Mediating Role of Employee Engagement in the Relationship between Flexible Work Arrangements and Turnover Intentions among Highly Educated Employees in the Republic of Serbia"

_behavsci, 2023, doi:10.3390/bs13020131_

Round 1

Reviewer 1 Report

I think that this study is a good attempt to investigate the relationship between flexible work arrangements and turnover intention with a focus on the mediating role of employee engagement. However, despite the attempt, there are seriously many holes to fill in as an empirical paper that tries to settle down in the research area of organizational behavior.

First, I’d like to point out that the abstract of the paper has to be greatly improved so that the potential readers can easily find the values of this study. “The challenges faced by modern organizations, such as globalization, digitalization, energy, economic, political, health (Covid-19 pandemic) and other crises, require organizations world- 10 wide to adapt business patterns and find adequate ways to keep and improved their position in the market. Organizations want to attract, motivate, and retain the best candidates, and accordingly use different strategies to succeed. One of the ways that companies can manage their human resources more successfully is through job redesign and the implementation of flexible work arrangements. (line 9-14)” This part is completely redundant and unnecessary to highlight the value of the study. Extremely boring introduction! Half of the abstract is valueless in terms of the convention of abstract of empirical paper in the research area of organizational behavior.

In abstract, the authors begins with very succinct introduction of their main research questions. Their arguments have to follow afterward. Simply saying, what their research hypotheses are have to appear. Then, to test their hypotheses, their efforts especially in data collection, analysis, and results have to be SUCCINCTLY described. What must appear after the authors show methods and results is that how their study is valuable in terms of the contributions to the existing literature. I would like to see such things in the abstract of this study. Therefore, I suggest the abstract must be completely rewritten to nicely respond to my comments. 

Too many acronyms appear without explaining what it exactly means. I have to ask the authors not to simply assume that the readers of their paper would be familiar with such mindlessly written acronyms. What would be FWA, TI, and EE? What the authors know is not always what the readers would understand. All the concepts must be introduced with full description of what the acronym refers to. 

The development of hypotheses must be all redone. I do not find the authors’ effort to develop their hypotheses with their good understanding about the existing literature. Citing the existing literature is not sufficient at all. As Sutton and Staw (1995) pointed out, the enumeration of existing literature is not a theory of an empirical paper. A set of hypotheses cannot be a theory proposed by the authors in a paper, only because some existing literature said so. There has to be something more that the authors attempt to add. Unfortunately, I do not find such new things in this paper. Every sing hypothesis has to be followed by one or two paragraphs in which the authors show why and who they come to propose the hypotheses. Therefore, I cannot help suggesting the authors to completely rewrite the section 2. Theoretical background. 

What is the population of this study? Unless the authors clearly define the definition of the population, the following procedure of collecting data is meaningless. To what population do the authors design the research? I’d like to clearly see the definition of population, and then see how the sample was chosen based on the population that the author aim for. Finally, I ask the authors to show how their sample is representative for their target population. Without my request clearly being addressed, the current study cannot be settled in the current outlet that the authors pursue for. 

All the variables, such as flexible work arrangement, turnover intention, and employee engagement, must be backed up by the existing literature in terms of how it measures in the body of paper. In the current manuscript, everything about the variable and the measure is ambiguous. I’d like to see the kind explanation from the authors. 

The analysis that the authors chose is path analysis that stems from structural equation modeling. I want the authors to know that the software they use is not equivalent with the analysis. Employing certain programs to the analysis does not confirm that the authors have appropriate knowledge about the analysis they use for their study. SmartPLS4 is not the same thing with PLS-SEM. I suggest that based on the firm foundation of path analysis in structural equation modeling, the authors kindly report every step they did for the analysis. I would like to point out that the presentation of result tables without proper description of how the analysis is done is nothing.

The principles about empirical study are all simple. I recommend the authors to refer to good existing papers that have settled down in prominent and prestigious journal in the research area of organizational behavior, such as Academy of Management Journal, Administrative Science Quarterly, Journal of Management, Journal of Applied Psychology, and the like. You may be easily able to find some good papers using similar topics and the exact methodology as your study. Using such papers as exemplar will help you greatly improve the quality of your paper. 

Good luck!

References

Sutton, R. I., & Staw, B. M. (1995). What theory is not. Administrative Science Quarterly, 371-384.

Author Response

Dear reviewer,

Thank you for the opportunity to improve our paper. After we analyzed your review, we made changes in all parts you suggested and tackled all the issues that were underdeveloped in previous version of the paper. All changes in the text are in blue color, so you will be able to find these parts easily. 

The  main changes are made in:

  1. the abstract - we rewrote the abstract and now it is shorter and more specific.
  2. the acronyms - we put the full titles of all three variables used in the research as flexible work arrangements, employee engagement and turnover intentions.
  3. the development of hypotheses - we redone this part of the paper totally, so you will se new section, total in blue color, with new perspective and new approach (we hope you will be satisfied with it).
  4. the population of this study - we explained the population of the study, and the approach how we get our sample. It is important to mention that we only had the official data from Serbian office for statistics, regarding the number of highly educated employees, and some speculation about the number of male and female (more female with high education degree than man).
  5. the variables - we explained how variables were measured and gave the basic explanation of the concept.
  6. the PLS-SEM - we explained in the new subsection 3.3. the procedure of PLS - SEM and all the steps we used in our analysis.

All changes are given in blue color in the new paper.

Best regards,

the authors.

Reviewer 2 Report

The subject addressed is interesting and important for the economic literature and for policy makers. After reading the paper, I find the analysis worth to be to be published. In order to improve the paper for publication, I would have the following mentions:

·         the title of the work should be revised, to be more concise and clear;

·         the first paragraph in the abstract requires the attention of the authors, the main idea is lost in the enumerations of the difficulties faced by modern organization, it should be reformulated;

·         in the econometric analysis, it was used a sample of 514 highly educated employees. The authors must explain how they chose this sample. As I read from the paper, the initial size of the sample was 582, the authors must explain why they took into account only the 514.

·         the methodological approach is clearly presented and appropriate to the subject addressed. The results are clearly presented and obviously useful for policy makers. Conclusions section is well-written. I would like to suggest the author(s) to present their conclusions compared with the other previously obtained for Serbia or other countries.

·         the article is written in a “clumsy” English and requires attention from the authors.

Author Response

Dear reviewer,

Thank you for the opportunity to improve our paper. We have made almost all changes you proposed regarding our paper. All the changes re given in blue color in sections related to your comments.

  1. the title of the work - we tried to make it shorter but the main idea is that, and we searched for some similar research, all of them have similar constructs: main variables, location and sample is mentioned in the title.
  2. the abstract - we have completely reformulated the abstract;
  3. in the econometric analysis, it was used a sample of 514 highly educated employees - we used only 514 because we got 582 responses, but after finding outliers and several incomplete responses, we excluded those and got 514 valid answers. This is explained in the sample subsection in the paper.

Thank you for your valuable comments.

Best regards,

the authors.

Round 2

Reviewer 1 Report

All my concerns in the previous manuscript are well addressed by the authors in the revised manuscript. Therefore, I give no further comments. 

Good job!

Reviewer 2 Report

yes, I agree to review the improved version of the paper.